# Inertial Sensors for Hip Arthroplasty Rehabilitation: A Scoping Review

**DOI:** 10.3390/s23115048

**Published:** 2023-05-25

**Authors:** Patricia Acosta-Vargas, Omar Flor, Belén Salvador-Acosta, Franyelit Suárez-Carreño, Marco Santórum, Santiago Solorzano, Luis Salvador-Ullauri

**Affiliations:** 1Facultad de Ingeniería y Ciencias Aplicadas, Universidad de Las Américas, Quito 170125, Ecuador; omar.flor@udla.edu.ec (O.F.); franyelit.suarez@udla.edu.ec (F.S.-C.); 2Intelligent and Interactive Systems Laboratory, Universidad de Las Américas, Quito 170125, Ecuador; maria.salvador.acosta@udla.edu.ec (B.S.-A.); santiago.solorzano@udla.edu.ec (S.S.); 3Facultad de Tecnologías de Información, Universidad Latina de Costa Rica, San José 11501, Costa Rica; 4Facultad de Medicina, Universidad de Las Américas, Quito 170125, Ecuador; 5Departamento de Informática y Ciencias de la Computación, Escuela Politécnica Nacional, Quito 170525, Ecuador; marco.santorum@epn.edu.ec; 6Department of Software and Computing Systems, University of Alicante, 03690 Alicante, Spain; lasu1@alu.ua.es

**Keywords:** arthroplasty, hip, inertial, rehabilitation, sensors, IMU, scoping review

## Abstract

The objective of this scoping review is to characterize the current panorama of inertia sensors for the rehabilitation of hip arthroplasty. In this context, the most widely used sensors are IMUs, which combine accelerometers and gyroscopes to measure acceleration and angular velocity in three axes. We found that data collected by the IMU sensors are used to analyze and detect any deviation from the normal to measure the position and movement of the hip joint. The main functions of inertial sensors are to measure various aspects of training, such as speed, acceleration, and body orientation. The reviewers extracted the most relevant articles published between 2010 and 2023 in the ACM Digital Library, PubMed, ScienceDirect, Scopus, and Web of Science. In this scoping review, the PRISMA-ScR checklist was used, and a Cohen’s kappa coefficient of 0.4866 was applied, implying moderate agreement between reviewers; 23 primary studies were extracted from a total of 681. In the future, it will be an excellent challenge for experts in inertial sensors with medical applications to provide access codes for other researchers, which will be one of the most critical trends in the advancement of applications of portable inertial sensors for biomechanics.

## 1. Introduction

Inertial devices, specifically inertial measurement units (IMUs) [1], have become an increasingly popular tool for objectively quantifying and evaluating human body motion in the healthcare industry. IMUs have been used in various applications [2] including measuring human body movements, postural sway, and anticipatory postural movements to diagnose and assess movement disorders.

However, IMUs have a significant disadvantage: they are generally affected by cumulative errors related to changes in the position of the human body [3]. To overcome this limitation, 3D technologies including depth cameras such as Kinect, Orbbec 3D, and other devices have been used to track physical rehabilitation in patients using non-invasive technology [3] that does not require the installation of sensors in the patients.

An inertial sensing unit consists of accelerometers, gyroscopes, and magnetometers. While a triaxial accelerometer can measure the proper linear acceleration of movements in a fixed three-dimensional (3D) frame per sensor, the estimated data includes components of motion and gravity. However, wearable sensors have not yet become standard in posturography [3] due to the unknown accuracy of IMU-based assessments for balance assessments. Portable sensors that measure balance would be ideal because of their low cost and ease of transport in different environments.

Currently, wearable inertial sensor technologies are used to analyze human movements, providing versatility and cost reduction compared to laboratory techniques. These technologies are applied in various fields, such as clinical, healthcare, sports, and textiles, to generate value-added products and services by interpreting human movement.

Additionally, Google Trends [4] examines the level of interest in searches for specific keywords over a given period. By analyzing samples of these searches, we can measure the level of interest in various topics, such as inertial sensors, IMUs, rehabilitation, and hip arthroplasty, across all Google searches performed over the past five years and compare the popularity of these searches.

As shown in Figure 1, the term “IMU” began to grow in December 2022, and “rehabilitation” from January 2023, which could indicate an increase in interest or relevance of these topics at this particular time. December 2022 also sees increased demand for IMUs in various applications, from robotics to virtual and augmented reality technology.

The growing interest in rehabilitation in January 2023 could be related to the growing concern for health and well-being, particularly after the COVID-19 pandemic [5]. There may be a greater interest in finding solutions and resources for rehabilitation in general.

It is important to note that search trends on Google Trends are not always indicative of global trends but can be influenced by many factors, including current events and changes in search behavior. Based on the data presented in Figure 1, it is evident that the most popular topics were IMU and rehabilitation, while no significant trends were observed for searches related to inertial sensors and hip arthroplasty.

In this context, future sensors and systems development trends and prospects focus on typical MEMS sensors for Internet of Things applications. Future sensor trends focus on intelligence and lower power consumption, and [6] future sensors, such as event-based sensors, address big data and human–machine issues. Artificial intelligence and virtual reality technologies using sensor nodes and their wave identification are presented as future trends for various scenarios.

This document is divided into several sections. In Section 2, readers are introduced to inertial sensors for hip arthroplasty rehabilitation. Section 3 explains the research method used in this scoping review. Section 4 provides an overview of the results of the analysis performed in this literature review. In Section 5, a discussion of the results and limitations of this research is presented. Finally, Section 6 outlines the assumptions made in this study and identifies areas for future research.

## 2. Background and Motivation

The innovation of the technique or technology of inertial sensors for hip rehabilitation lies in their ability to provide objective and quantitative information about the patient’s movement and position during rehabilitation, which can help improve the effectiveness and efficiency of the rehabilitation process.

Unlike traditional rehabilitation techniques, which rely on the subjective evaluation of the therapist, inertial sensors can provide precise and objective measurements of the patient’s movements, allowing for more accurate tracking of progress and more efficient adaptation of therapies. Additionally, inertial sensors are portable and non-invasive, making them more comfortable and convenient for the patient.

A systematic review is necessary to evaluate the validity and reliability of inertial sensors for hip rehabilitation and identify the trends and limitations of their use. This review allows researchers and medical professionals to make informed decisions about the selection and use of inertial sensors for hip rehabilitation, which can significantly improve outcomes and treatment efficacy. Furthermore, this systematic review can also help identify gaps in existing research and highlight areas for future investigation.

Hip arthroplasty is surgery to replace a damaged hip joint with an artificial joint. The goal of this surgery is to relieve pain and improve the motor function of the joint. Hip arthroplasty surgery can be partial or total. Partial hip arthroplasty replaces only the head of the thigh bone, while total hip arthroplasty replaces the head of the thigh bone and the hip joint.

Figure 2 shows a bibliometric review considered in this study with VOSviewer [7], a computer program used for bibliometric analysis. This review focused on identifying and analyzing the terms related to the technologies used and inertial sensors for measuring and monitoring the implant provided in arthroplasty treatments. The resulting graph shows the co-occurrence of these terms in the scientific literature, with larger nodes indicating more frequently occurring times and stronger relationships between the words.

It is important to note that the limited amount of scientific literature on these developments influenced the results obtained in the bibliometric analysis. The lack of data on this topic may also limit the generalizability of the findings and the ability to draw broader conclusions. Nonetheless, this bibliometric review provides valuable insight into the current research on using inertial sensors in arthroplasty treatments and can help guide future research in this area.

The technologies shown in Figure 3 allow the study of postures, kinematics, pelvis position, and walking patterns using sensors adapted to total hip replacement cases. The use of IMU inertial sensors is related to technologies such as machine learning, support vector machine, sound feedback, and visual monitoring. Figure 4 presents the layout of some of the methods identified in this systematic review.

Figure 3 presents the elements that comprise each type of implant position measurement system incorporated in the complete hip arthroplasty process.

The five inertial sensor systems in Figure 2 have specific tasks related to measuring the prosthesis position. The work in [4] presents an encapsulated, single-position sensor at the back of the waist and hip level. The task of this wireless sensor is to transmit the tilt angles of the hip portion near the sensor to compare them with standard gait patterns and know the progress of the patient’s rehabilitation status. The development of [9,10] presents a system similar to the previous case but also places sensors on the patient’s legs near the upper thigh, knees, and ankles of both legs. The positions of the lower limbs and the hip position allow the specialist to know if abnormal situations are in the usual walking process.

The document [11] presents a system adapted to an artificial patella implanted in the femur’s upper part and serves as a base to maintain the proper position of the hip implant. It includes in its structure force and inclination sensors that allow transmission data to the computer, enabling the specialist to know the load to which these artificial kneecaps are subjected and to have evidence of the state of movement and forces that the patient generates in his rehabilitation.

The study [12] uses a system similar to that of Figure 3b; the system is based on determining the moment of contact with the ground and the inclination of the knee. This system measures the gait time and thus generates sounds that will be fed back to the patient who tries to align his movements to the proposed sound, according to the footsteps and the typical gait pattern to which he should be moving. The optical tracking system proposed in [13] uses markers in the positions specified in Figure 3e, and using vision systems allows one to identify the places reached by the hip and limbs of the person in his rehabilitation process.

The technologies enabling Figure 3 to function effectively include three main areas, including inertial sensors that wirelessly transmit angular values and accelerations according to the movement of the hip and lower limb joints. A second technology coordinates the sound of normal gait so that the treated patient knows at what pace to move and, during therapy, allows a regular speed to be set to match the typical gait pattern.

The third technology incorporates visual sensors similar to Kinect systems in which the limbs are detected through spheres strategically placed as markers that indicate to the vision system the position of the limbs and movement speeds. This system will compare this movement with standardized movements and thus inform the specialist about the patient’s mobility status.

## 3. Materials and Methods

A scoping review (SR) [11] was initiated to conduct this study, which began by establishing a review protocol, research question, and methods. The PRISMA extension for scoping reviews (PRISMA-ScR) [14] checklist was utilized, which includes twenty essential information elements and two optional elements that are typically used in health-related research [15].

The research question is as follows: How can inertial sensors be used in hip arthroplasty rehabilitation, and what benefits and limitations are associated with their use? To avoid biases, we adapted the PRISMA-ScR checklist in this study to identify studies on applying inertial sensors in hip arthroplasty. Appendix A presents a list proposed by PRISMA-ScR, which indicates the number of pages that meet or do not meet the twenty-two aspects described in the seven sections of the SR, including the title, abstract, introduction, methods, results, discussion, and financing.

This review was conducted in five phases, including defining research questions to establish the scope and developing a search strategy to retrieve all relevant papers, screening the articles to identify the most appropriate ones, coding the documents that meet the classification structure, and performing a data extraction and review procedure to obtain the results.

### 3.1. Definition of the Research Questions to Determine the Scope

Our primary research question is: How can inertial sensors be used in hip arthroplasty rehabilitation, and what benefits and limitations are associated with their use?

The research question is relevant as it addresses a topic of great importance in post-arthroplasty hip rehabilitation. Using inertial sensors in post-arthroplasty hip rehabilitation may provide a non-invasive and accurate way to monitor patient movement and position. This research may help develop personalized therapies and improve treatment efficacy.

In addition, the research question also addresses the limitations of inertial sensors, which may help researchers and medical professionals identify areas of improvement and develop solutions to overcome these limitations. From this question related to inertial sensors, we present three objectives.

The first objective of this research is to present information related to the most relevant research on IMU applied in arthroplasty from 2010 to 2023. This systematic review (SR) contains several articles extracted from digital libraries, specifying authors, year of publication, and SCImago Journal Rank (SJR).

The second objective is to identify different IMU technologies and determine trends.

The third objective is to determine the main functions of IMUs and their technological development.

Our research examines the results of primary published studies on IMUs and their application in hip arthroplasty to identify the current advantages and limitations of IMUs and their development.

The research questions allow us to identify the advantages, limitations, and remaining issues in this area and are defined as follows:

RQ1. What are the most common types of IMUs used to assess movement in hip arthroplasty? This question investigates the types of IMUs taken into account when evaluating movements in hip arthroplasty from studies between 2010 and 2023.

RQ2. What are the main functions performed by IMUs to assess movements in people who have undergone hip arthroplasty? This question aims to classify the functions and techniques used to assess movements in people who have undergone hip arthroplasty.

RQ3. What methods are applied in IMUs to evaluate movements in people who have undergone hip arthroplasty? This question determines the methods applied in IMUs to evaluate the movements of people who have undergone hip arthroplasty.

RQ4. What types of research and contributions have been found related to IMUs for evaluating the movements of people who have undergone hip arthroplasty? This question distinguishes the type of research and contributions related to IMUs in assessing the movements of people who have undergone hip arthroplasty.

RQ5. What advantages have been found in the application of IMUs? This question classifies the main advantages of using IMUs in hip arthroplasty.

RQ6. What limitations have been identified in the application of inertial sensors? This question seeks to determine the main limitations of using inertial sensors in hip arthroplasty.

### 3.2. Search Strategy to Extract the Documents

A series of queries were conducted based on the research questions to obtain the primary research for this scoping review. The articles’ quality and relevance, population, intervention, comparison, and outcome (PICO) criteria were applied [9]. The population criteria referred to published studies, while the intervention criteria focused on using inertial sensors in hip arthroplasty. The comparison criteria required carefully selected studies involving inertial sensors and the type of research. The outcome criteria involved published studies on inertial sensors and hip arthroplasty. Taking into account the quality of the extracted articles, five new questions were formulated based on the PICO criteria and are presented in Table 1.

The search was conducted in October 2022; we selected five academic research databases used in engineering and healthcare to retrieve primary information: ACM Digital Library, PubMed, Science Direct, Scopus, and Web of Science (WOS).

The search queries for each database were created using Boolean operators to combine related search words. Table 2 displays the selected databases, query string used, and some extracted documents. The databases included Scopus- and WOS-indexed databases, which are well-known citation databases, as well as peer-reviewed literature abstracts, scientific journals, books, and conference proceedings. The query string was applied to the publication title and abstract using keywords such as “IMU”, “sensors”, “inertial”, “arthroplasty”, “hip”, and “telerehabilitation”. The same search syntax was used for all five databases, and the search period was set to include studies published between 2010 and 2023.

### 3.3. Screening of Documents

This statement indicates that the evaluators who applied the inclusion and exclusion criteria for selecting the primary documents had a moderate agreement. Cohen’s kappa coefficient is a statistical measure used to assess inter-rater reliability or the degree of agreement between two or more evaluators. The coefficient ranges from −1 to 1, with values closer to 1 indicating higher agreement between evaluators. In this case, a coefficient of 0.4866 indicates moderate agreement among the evaluators. A percentage of 97.4% indicates a moderate agreement of the evaluators with the inclusion and exclusion of documents. This statement outlines the inclusion and exclusion criteria used to select primary studies for this systematic review. The inclusion criteria require that the preliminary research is (1) published in journals, conferences, books, or book chapters on inertial sensors and hip arthroplasty between 2010 and 2023; (2) a peer-reviewed primary study; and (3) written in the English language. These criteria help ensure that the selected studies are relevant, up-to-date, and highly qualified.

The exclusion criteria specify that the preliminary study should not be related to (1) a summary of the central discourse, an introduction to the workshop, or only an abstract; (2) duplicate papers of the same research from different sources; and (3) secondary studies related to literature reviews. These criteria help exclude studies that do not contain sufficient data, redundant or repetitive studies, and studies that are not primary research but secondary analyses or literature reviews.

Overall, these criteria help ensure that the selected studies are relevant to the research question and are of high quality while excluding studies that are not primary research or contain insufficient data.

The PRISMA-ScR [14] tool is applied for conducting systematic reviews. PRISMA-ScR stands for preferred reporting items for systematic reviews and meta-analyses extension for scoping reviews and is a set of guidelines for reporting systematic and scoping reviews. By following these guidelines, the review authors can improve the quality and clarity of the review and avoid bias in the selection of studies.

The review and selection process is presented in a flowchart, which provides a clear visual representation of the review process. The flowchart includes the different phases of the review, such as identification, screening, eligibility, and inclusion, as well as the number of articles found and the number of articles rejected for different reasons. This level of detail helps ensure transparency and reproducibility of the review process, allowing others to assess the rigor of the review and potentially replicate it.

The statement mentions that the number of articles that met the inclusion requirements for the study is extracted. This indicates that the review authors used a systematic and rigorous approach to selecting primary studies, which helps ensure that the review is comprehensive and informative.

This statement refers to Figure 4, which illustrates the PRISMA flowchart [14], a standardized tool used to guide systematic reviews and meta-analyses. The flowchart visually represents the four phases of the review process: identification, screening, eligibility, and inclusion.

Phase 1 of the research involved the identification of relevant articles related to the research topic. The authors searched multiple databases, including ACM Digital Library, PubMed, ScienceDirect, Scopus, and WOS.

After conducting the searches, the authors identified a total of 681 articles. Specifically, they found 8 papers from ACM Digital Library, 65 articles from PubMed, 62 from ScienceDirect, 543 from Scopus, and 3 from WOS.

The identification phase is essential in any research study, as it helps researchers gather relevant information and literature related to the research topic. By searching multiple databases, the authors collected a comprehensive set of articles that can be used to inform the subsequent phases of their research.

Phase 2 of the investigation involved the selection of the articles identified according to the inclusion and exclusion criteria.

Initially, the authors found a total of 681 articles from multiple databases. After applying the inclusion and exclusion criteria, 59 articles were excluded because they were duplicated in different databases, leaving 622 articles.

Next, the authors reviewed all 622 articles and excluded 571 articles unrelated to inertial sensors and hip arthroplasties, such as literature reviews and other unrelated topics, leaving 51 articles for the next phase. Then, 28 articles that were not in the English language or that included abstracts were excluded. This left the authors with 23 articles that were relevant to the research topic and met the inclusion criteria.

The selection phase is crucial as it helps reduce the number of articles to those most relevant and valuable for the study. By applying strict inclusion and exclusion criteria, the authors could select only the most relevant articles for the next phase of their research.

Phase 3 of the research involved an in-depth full-text review of the 23 articles identified in the previous phase.

The authors thoroughly reviewed each article to determine its relevance to their inertial sensors and hip arthroplasty research topic. They focused on primary studies that explicitly addressed this topic.

After the full-text review, the authors did not exclude any of the 23 articles. All articles met the eligibility criteria and provided relevant information for the research study.

The eligibility phase is essential as it allows the authors further to assess the selected articles’ quality and relevance. By conducting a thorough full-text review, the authors could ensure that the selected articles were appropriate for their study and would provide valuable insights into their research topic.

Phase 4: Inclusion. The authors analyzed a total of 23 full-text documents for quantitative synthesis. Cohen’s [16] kappa coefficient, which represents the degree of accuracy and reliability in ranking the selected articles, was also applied. Cohen’s kappa was 0.4866, representing moderate agreement among the evaluators. The 23 articles answering the research questions on inertial sensors applied in hip arthroplasty detailed in Table 1 were evaluated for quality. This quality assessment (QA) aims to weigh the importance of each selected article to argue the results and guide the analysis.

### 3.4. Data Extraction and Review Process

The extraction of primary articles was an iterative process and was separated into stages, with different events carried out at each stage.

The authors used multiple databases to identify relevant articles for their study, including ACM Digital Library, ScienceDirect, Scopus, PubMed, and WOS. The data from ACM Digital Library were exported in BibTeX (BIB) format, while the other databases were transferred in Research Information Systems (RIS) format.

The authors then imported the information from all five databases into the StartLapes version 2.3.4.2 tool. This tool automatically removed duplicate research from the extracted databases, ensuring that the authors did not include duplicate studies in their analysis.

The four phases described in the PRISMA flowchart [17], which is a widely recognized tool for conducting and reporting systematic reviews and meta-analyses, were applied in this process. These phases include identification, screening, eligibility, and inclusion of relevant studies.

In the final phase of the research, the authors took the 23 selected articles and analyzed them in Microsoft Excel.

Table 3 contains the selected articles, with the results of quality control. It includes the ID, the name of the publication, the five quality questions, the applied score, and the normalization. The normalization column uses a standard scale that goes from 0 to 1.

To calculate the normalization values [18], the authors used Equation (1) as follows:(1)$$Normalization=\frac{Score-minimum(Score)}{\left[maximum(Score)-minimum(Score)\right]}$$

The score is the value obtained for each quality question, the minimum score is the lowest among all selected articles, and the maximum score is the highest value obtained for that question among all selected articles.

The minimum (score) equals 0, the maximum (score) is equal to 5, and the score is the value considered in Table 3.

## 4. Results

In this section, the authors answered two of the six research questions posed in defining the research questions to determine the scope section. These questions were:

(1) What is the bibliometric assessment to collect yearly publication records on the increase in research over time from journals, conferences, books, and book chapters on inertial sensors and hip arthroplasty?

(2) What is the scoping review to map studies according to perceptions of inertial sensors and hip arthroplasty?

For the first question, the authors conducted a bibliometric assessment to collect yearly publication records on the increase in research over time from journals, conferences, books, and book chapters on inertial sensors and hip arthroplasty.

For the second question, the authors conducted a scoping review to map studies according to perceptions of inertial sensors and hip arthroplasty. They used a systematic approach to identify and select relevant studies from multiple databases. They applied inclusion and exclusion criteria and performed a thorough full-text review of the selected articles.

Through this scoping review, the authors could map out the current research on inertial sensors and hip arthroplasty, identifying key themes and areas of focus. They were able to provide valuable insights into the current state of the research in this area and help guide future research efforts.

### 4.1. Bibliometric Analysis

This analysis aims to answer RQ1. Figure 5 shows the evolution of scientific production by presenting the number of papers published each year. The year with the highest scientific production on inertial sensors and hip arthroplasty was 2019, followed by 2021. We could not find any articles related to this topic between 2010–2012 and 2014. However, in 2013, 2015, and 2018, publications accounted for 4.35% of each year. In contrast, the percentage of publications increased to 8.70% for 2016, 2017, 2020, 2022, and 2023. The year 2021 had 17.39% of the publications, and 2019 had the highest percentage of 26.09%, representing the most significant scientific production on inertial sensors for hip arthroplasty.

Figure 6 shows the number of studies found in the databases; 13% corresponds to the PubMed database with three studies, 82.6% in the Scopus-indexed database with two articles, and 4.4% in the Science Direct database with one article. The most significant research in this literature review is focused on journals. The most significant number of scientific papers are in PubMed.

### 4.2. Review of the Literature to Map the Studies

Table 4 presents the 23 primary studies selected based on their most recent year of publication. Each study in the table includes the following information: (1) article number, (2) an assigned indicator generated from the first author’s name, consisting of the first two letters of their last name and the first letter of their first name, followed by the year of publication, (3) the title of the scientific article, (4) reference, and (5) year of publication.

To answer our research questions, we thoroughly analyzed these 23 primary studies, examining sections such as abstracts, keywords, introductions, methodologies, discussions, and conclusions. We documented the relevant characteristics of each primary study in a spreadsheet for further reference.

By focusing on these six research questions, we extracted the essential characteristics of each primary study and systematically analyzed their findings.

In the following, we present and discuss the answers to the research questions of this study; the dataset and evaluation for replication are available in the Mendeley repository [38].

By analyzing the 23 scientific articles, we highlight and identify in each study the type of sensor used for hip arthroplasty, the main functions, the methods applied, the research related to the study, the advantages and limitations that answer the six research questions raised in this scoping review. These results are summarized in Table 5.

#### 4.2.1. RQ1—What Are the Most Common Types of IMUs Used to Assess Movement in Hip Arthroplasty?

In this question, we answer the types of inertial sensors taken into account in assessing motion in hip arthroplasty from studies conducted between 2010 and 2023. The authors [21,23] argue that the most commonly used inertial sensors for assessing motion in hip arthroplasty are the IMU (inertial measurement unit).

These sensors combine accelerometers and gyroscopes to measure acceleration and angular velocity in three axes. Refs. [8,22] indicate that an IMU-based intelligent hip testing system was used to measure the orientation and velocity of hip motion.

Another article by [9] argues that the most commonly used inertial sensors for assessing motion in hip arthroplasty are IMUs and acceleration sensors.

Ref. [29] refutes that force, position, and motion sensors are the inertial sensors most commonly considered for assessing motions in hip arthroplasty. The Force-PRO device combines all three types of sensors to measure the hip implant’s force, position, and motion during the surgical procedure and patient recovery.

Ref. [12] replicates that the data collected by these IMU sensors are used to analyze gait and detect deviations from normal.

Ref. [13] argues that IMU sensors are used to measure the position and motion of the hip joint and assess the quality of surgery and patient recovery.

Of the 23 studies analyzed, 95.7% use IMU inertial sensors; Ref. [29] is the only one that applies Force-PRO IMU, a high-precision inertial measurement unit (IMU) developed by Force-Technology. This unit combines an accelerometer, gyroscope, and magnetometer to provide high-quality, accurate inertial measurement data.

#### 4.2.2. RQ2—What Are the Main Functions Performed by Inertial Sensors to Assess Movements in People Who Have Undergone Hip Arthroplasty?

In this question, we classify the functions and techniques used to assess the movements of people undergoing hip arthroplasty. We found that the studies [8,12,13,22,23] argue that the main functions of inertial sensors are capable of measuring various aspects of motion, such as velocity, acceleration, and body orientation. A second paper [9] discusses the main functions performed by inertial sensors in patients who have undergone hip arthroplasty, such as (1) the measurement of gait kinematics and (2) the classification of gait in inertial sensors as normal gait, toe walking, heel walking, and assessment of balance during gait. Refs. [9,13,21,22,23,29,31,33,37] indicate that the functions allow measurement of the force at the hip joint during movement, evaluation of the implant position, and measurement of the velocity and acceleration of movement in three axes, as well as analysis of the movement patterns to detect any abnormal deviations.

Refs. [20,27,28] relate to IMU repeatability in hip replacement, post-surgery gait analysis, and gait classification input representations. The studies [8,12,13,22] in question are related to aspects such as movement, speed, acceleration, orientation, monitoring the movement of the implant in 3D, and movement in search of abnormal deviations.

Research [24,32] addresses issues related to the measurement of the angle of the prosthesis in real-time; Refs. [25,26] focus on the follow-up of rehabilitation through DCNN and distance training.

Research [19] addresses the follow-up of dual-motion hip implants, while [30] argues that a hip replacement affects body movements; [34] focuses on open-source surgical navigation using optical tracking; [35,37] presents improved results of hip arthroplasty using the inertial measurement unit (IMU); and [36] uses Smart Trail to estimate hip movement.

When analyzing the most frequently repeated keywords in the twenty-three selected studies, we identified hip with a frequency of eleven times, followed by measurement and arthroplasty with six, and gait with five; summarized in Figure 7 in a word cloud.

#### 4.2.3. RQ3—What Methods Are Applied in Inertial Sensors to Assess Movements in People Who Have Undergone Hip Arthroplasty?

In this scoping review, we found methods applied with inertial sensors to assess the movements of people who have undergone hip arthroplasty. The methods used in inertial sensors to assess the movements of persons undergoing hip arthroplasty according to [13,23,29] include (1) the support vector machine type trained with gait data based on inertial measurement units; (2) an intelligent hip testing system, a fixed pelvis model, angle analysis, and motion testing to assess movements; (3) data collected to calculate gait characteristics, stance time, and gait speed; and (4) an IMU system that is accurate and reliable for assessing hip posture after total hip arthroplasty. Refs. [9,12,21] argue that (1) the data collected is used to analyze gait symmetry, hip joint range of motion, and other important clinical indicators related to recovery after surgery. (2) It is mentioned that dual sonification is used to help the patient improve gait rhythm and symmetry. (3) An optical tracking system collects data. Finally, [8,22] indicate that among the methods used, we have (1) an intelligent system of hip tests, angle analysis, and movement tests. (2) The data allows gait characteristics, support time, and speed to be calculated to classify gait into different categories. (3) An intelligent hip testing system for angle analysis and motion testing. Refs. [20,27,28] address the creation of a reliable IMU for measuring hip mobility, gait measurement, and gait classification. Refs. [31,32,33,37] analyze aspects related to speed, precise measurement of the position of the prosthesis, position estimation using IMU, and data recorded by the camera, as well as the center of the joint, which have been validated with an optical measurement system. Refs. [24,25,26] advocate accurate classification, real-time feedback, and monitoring for effective rehabilitation using remote rehabilitation approaches and technologies. Ref. [19] focuses on the inertial tracking of dual-motion hip implants; [30] discusses wearable sensors for monitoring lower body movements. On the other hand, [34] offers an affordable PAO navigation solution, [35] discusses the use of the Kalman filter to improve the accuracy of the sensor, and finally, [36] proposes the Smart Trail system to predict movements from the hip.

#### 4.2.4. RQ4—What Type of Research and Contributions Has Been Found Related to Inertial Sensors for Assessing the Motions of People Who Have Undergone Hip Arthroplasty?

With this question, we answer the type of research and contributions according to [13,22,23]; we found research contributing to (1) the development of a machine learning algorithm based on a single-class support vector machine (SVM). (2) IMUs are an accurate and reliable tool to assess people’s movements after hip arthroplasty. Refs. [8,29] suggest that (1) IMU-based systems help to monitor and evaluate the rehabilitation of patients after hip arthroplasty. (2) Inertial sensors are applicable for gait classification. (3) The Force-PRO device investigates how different factors, such as implant type and rehabilitation protocol, affect implant loading and range of motion after hip arthroplasty. Refs. [9,12,21] argue that (1) inertial sensors are useful and applicable tools for gait classification. (2) The sensor data produces a sound representing the patient’s movement that can help improve their gait technique.

Refs. [20,27,28] suggest that IMUs help evaluate THA, gait, and hip arthroplasty and provide data analysis with clinical applications. Refs. [31,33,37] support the use of inertial sensors to achieve more precise orthopedic surgery, as they provide reliable results in prosthetics and surgeries. Refs. [24,32] show that improving precision, efficiency, and angles allow for quantifying the results and optimizing the precision of THR surgery. Refs. [25,26] highlight the improvement in precision, efficiency, measurements, outcomes, and personalized rehabilitation that can be achieved through its approach. Ref. [19] focuses on inertial sensors to monitor hip arthroplasty, while [30,34,35,36] focus on reach, mobility, sensors, improvement of precision, safety, quantitative results, and rehabilitation.

#### 4.2.5. RQ5—What Advantages Were Found in the Application of Inertial Sensors?

In this scoping review, according to [12,21,23], we found the following advantages of applying inertial sensors in hip arthroplasty. (1) Collect gait data in a real-world environment without requiring a specialized laboratory. (2) Analyze gait from different perspectives using motion parameters. (3) Provide a visual explanation of detected pathological gait patterns through a graphical representation of motion parameters. (4) Other methods offer greater accuracy, portability, versatility, and efficiency.

Refs. [9,13,29] found the following advantages. (1) They are small, light and inexpensive, ideal for portable feedback systems. (2) They provide motion information such as velocity, acceleration, and joint angles. (3) They are easy to use and helpful for monitoring recovery after hip arthroplasty. (4) They provide accurate real-time measurements. (5) They allow for a better review of the patient’s evolution and recovery, with less invasiveness than other techniques. Finally, [8,22] argue that the benefits are (1) the ability to measure hip posture non-invasively and continuously is a great advantage. (2) The device’s ease of use and portability allow measurement in a clinical setting or the patient’s home. (3) The ability to provide real-time information. Refs. [20,27,28] have shown that IMUs offer advantages such as portability, real-time, precision, and reliability. Refs. [19,24,25,26,30,31,32,33,34,35,36,37] have shown that IMUs for hip arthroplasty offer precision, safety, low cost, and relatively low complexity.

#### 4.2.6. RQ6—What Disadvantages Were Found in the Application of Inertial Sensors?

According to [9,12,21,23], among the disadvantages are (1) a lack of precision in the measurement of joint angles. (2) The need for periodic pre-calibration to ensure data accuracy. (3) Limitations in accuracy, stability, measurement range, orientation sensitivity, and cost. Refs. [8,13,22,29] examining inertial sensors found the following disadvantages: (1) the need for additional processing to remove noise and biases in the inertial sensor data. (2) The difficulty in obtaining an accurate representation of physical activity in case of complex movements or sudden changes in velocity. (3) The dependence on proper sensor placement to obtain accurate measurements is sensitive to unwanted body movements, which can affect measurement accuracy. Refs. [19,20,24,25,26,27,28,30,31,32,33,34,35,36,37] have revealed limitations in inertial sensors used in hip arthroplasty, such as noise, interference, inaccurate joint angle measurement, problems with training data, lack of sensitivity, and temperature variation in small samples.

## 5. Discussion

This scoping review was derived from the PRISMA-ScR method [14] to contribute to the quality of the evaluated content and the clarity of the subject, preventing bias. This process consists of filtering in three fundamental steps, which allows debugging guidance for reviewers. In addition, the methodology used included a list of progressive elements described in scenario A, where the information refinement to improve quality is observed. It was observed in the review that most of the authors contribute to the following:

Patient populations, research designs, and sensor protocols varied between studies, as shown in Figure 5, noting notable growth in 2019 and 2021, Regarding the most common types of sensors [8,13,21,23] to evaluate hip arthroplasty movements, the most widely used are IMU sensors, which combine accelerometers and gyroscopes to measure acceleration and angular velocity in three axes. Postoperative treatment is envisioned to improve the quality of life in patients with hip fractures, including rehabilitation strategies with inertial sensors.

Wearable inertial sensors offer an accessible tool to support our understanding of the use of inertial sensors for hip arthroplasty rehabilitation and postoperative treatment. Different studies were identified that used portable inertial devices to evaluate hip arthroplasty rehabilitation. An IMU-based intelligent hip test system can be used to measure the orientation and speed of hip movement. However, the most widely used inertia sensors to assess movement in hip arthroplasty are IMUs and acceleration sensors.

In [22,29], it is justified that the inertial sensors most commonly considered to evaluate movements in hip arthroplasty include force, position, and movement sensors. The Force-PRO device combines all three sensors to measure the hip implant’s force, position, and movement during the surgical procedure and patient recovery.

Traditionally, in physiotherapy, joint mobility range evaluations have been based on the examination of conventional goniometers; there are also electrogoniometers and inclinometers, which give greater precision in measuring joint range. These devices facilitate the storage of information that can be recorded in a database for later interpretation, adding each record and giving the possibility of a more effective and natural movement identification.

Despite the advantages of the electrogoniometer, it cannot record additional information related to acceleration and speed in movements developed in specific planes and axes, variables that can be recorded by IMUs in static positions, dynamic, and during the execution of functional activities.

Some authors argue [9,12,21] that the data collected by the IMU sensors are used to analyze and detect any deviation from the normal and argue that the IMU sensors are used to measure the position and movement of the hip joint and to evaluate the quality of the surgery and patient recovery.

The main functions performed by inertial sensors to assess movement in people who have undergone hip arthroplasty, according to several authors, are to measure various aspects of movement, such as velocity, acceleration, and body orientation. Explicitly, the measurement of gait kinematics and the classification of the gait in inertial sensors in normal gait, on tiptoe, on heels, and the evaluation of balance.

Other authors [9,13] indicate that the sensors allow the measurement of the force in the hip joint during movement, the evaluation of the position of the implant, and the measurement of the speed and acceleration of the movement in three axes, as well as the analysis of movement patterns to detect abnormal deviations.

Regarding the advantage of the weakness of the inertial sensor, it is essential to note that a weak inertial sensor may provide less accurate and reliable measurements compared to a robust inertial sensor. However, in some cases, this weakness can be an advantage. For example, a weak inertia sensor may capture smooth and fluid movements better than fast and jerky movements in human motion monitoring. In addition, a weak inertial sensor may be less expensive and lighter, making it more suitable for portable and long-term use applications.

Regarding the methods applied to inertial sensors to evaluate the movements of people who have undergone hip arthroplasty, the review showed that the accelerometer and IMU are the most used technologies within the systems.

With the proposed methodology, the study shows that open-access publications and open-source software repositories present important models that can support growth, replication, reproduction, consistency, and inclusion in research. We found that 100% of the studies identified in this review were open-access. However, no studies provided open-source software attached to the articles. In the future, it will be an excellent challenge for experts in inertial sensors with medical applications to provide access codes for other researchers, which will be one of the most critical trends in advancing wearable inertial sensor applications for biomechanics.

The selected studies revealed that optoelectronic movements could be studied as a standard measurement tool in biomechanics, noting that IMU systems can perfectly measure the joint angle. Another strength of this review was applying Cohen’s kappa coefficient to achieve precision and reliability in selecting articles, with a moderate agreement of 97.4% between reviewers. On the other hand, a limitation of this study is that only open-access studies in English were included, and there may be possible debates of interest in other countries with different types of developments in the area of hip arthroplasty rehabilitation.

The authors propose a guideline for using inertial sensors (IMUs) for hip arthroplasty rehabilitation. Selecting suitable inertial sensors is essential to ensure accurate and reliable measurements. Arbitration, sampling rate, size, and battery life should be considered when selecting inertial sensors.

The inertial sensors must be placed correctly and securely on the patient to ensure accurate and reliable measurements. Sensors should be securely attached to the patient and placed strategically to capture relevant hip position and motion.

They are assessing the quality of measurements obtained by inertial sensors. Researchers and medical professionals must assess the measurements’ accuracy, repeatability, and validity to ensure their reliability.

The data processed by inertial sensors must be processed appropriately to obtain helpful information about the patient’s movement and position. This process may include the fusion of data from different types of sensors to provide a more complete and accurate measurement. It can be used to develop personalized therapies for the patient. Medical professionals can use the data to tailor treatment to the patient’s needs and assess rehabilitation progress.

Progress evaluation to assess the patient’s progress in hip rehabilitation after arthroplasty can be performed. Researchers and medical professionals can use the data to assess the efficacy and adjust the treatment plan. Patient safety involves ensuring that using inertial sensors is safe for the patient. Sensors must be correctly and securely placed to avoid injury or discomfort to the patient.

## 6. Conclusions

This review evidenced the current trends and developments related to inertial sensors in rehabilitating hip arthroplasty through primary scientific articles published in the last three years. The methodological nature did not include a deep analysis of the selected material; however, future research will include a systematic and extensive review of the different developments in hip arthroplasty rehabilitation, including contributions in open codes, intelligent tools, and force sensors. This research has managed to identify the contributions of inertial sensors to medical applications in rehabilitation. However, there is still much work to be completed so that researchers can develop tools with more significant technological potential, which involve intelligent elements for more significant results. In the same way, this research offers academics, researchers, and health professionals a vision of instrumentation resources, in this case, sensors, that can be useful in engineering developments applied to medicine.

In future projects, we will apply inertial sensor technology for personalized patient therapy in a web-based telerehabilitation system.

Finally, we recommend (1) continuing with the reviews to complement the evaluations by considering a heuristic method, which allows the generation of engineering contributions for hip arthroplasty rehabilitation with greater efficiency, (2) using this study as a starting point for the creation of new developments in the area of sensors that contribute to the solution of health problems, and (3) including the study of intelligent tools that contribute to medical developments with inertial sensors, which involve individualized aspects of the patient that provide integration to his health problem.

## Figures and Tables

**Figure 1 sensors-23-05048-f001:**
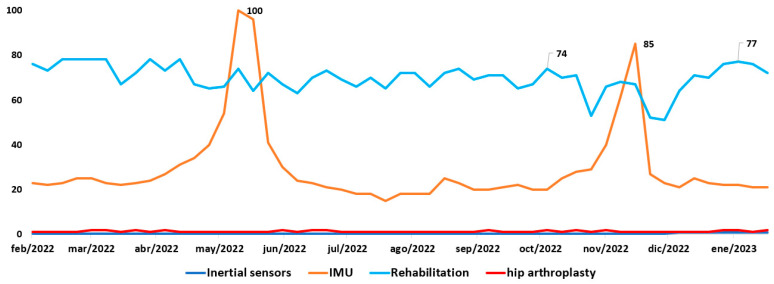
Google Trends is used to search for trends related to specific keywords over the period spanning from 2022 to 2023.

**Figure 2 sensors-23-05048-f002:**
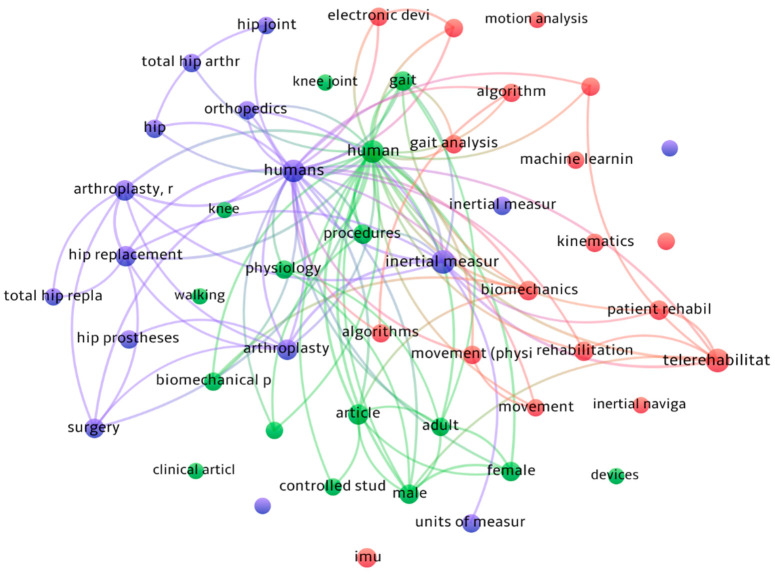
Bibliometric review based on the articles considered in this study.

**Figure 3 sensors-23-05048-f003:**
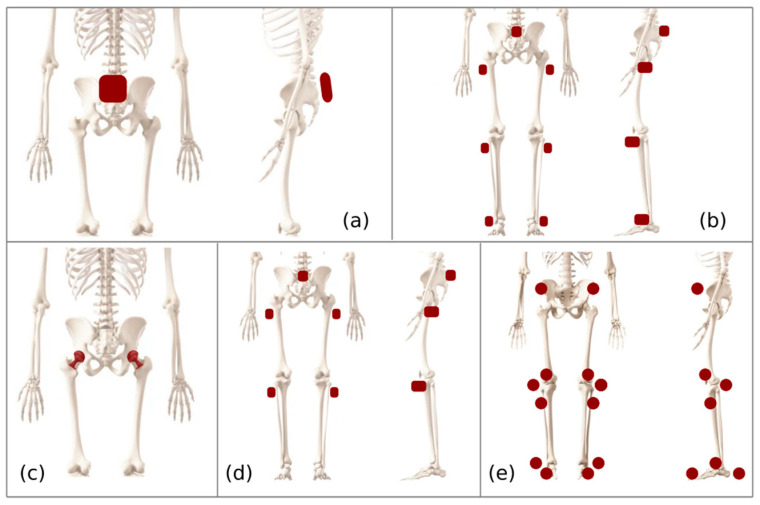
The component layout of multiple inertial sensors used in arthroplasty treatments for hip rehabilitation: (**a**) used in intraoperative monitoring [8]; (**b**) the use of SVM [9] and gait pattern detection [10]; (**c**) used to assess hip implant force and angle [11]; (**d**) used with sound feedback on gait; [12]; (**e**) optical motion analysis [13].

**Figure 4 sensors-23-05048-f004:**
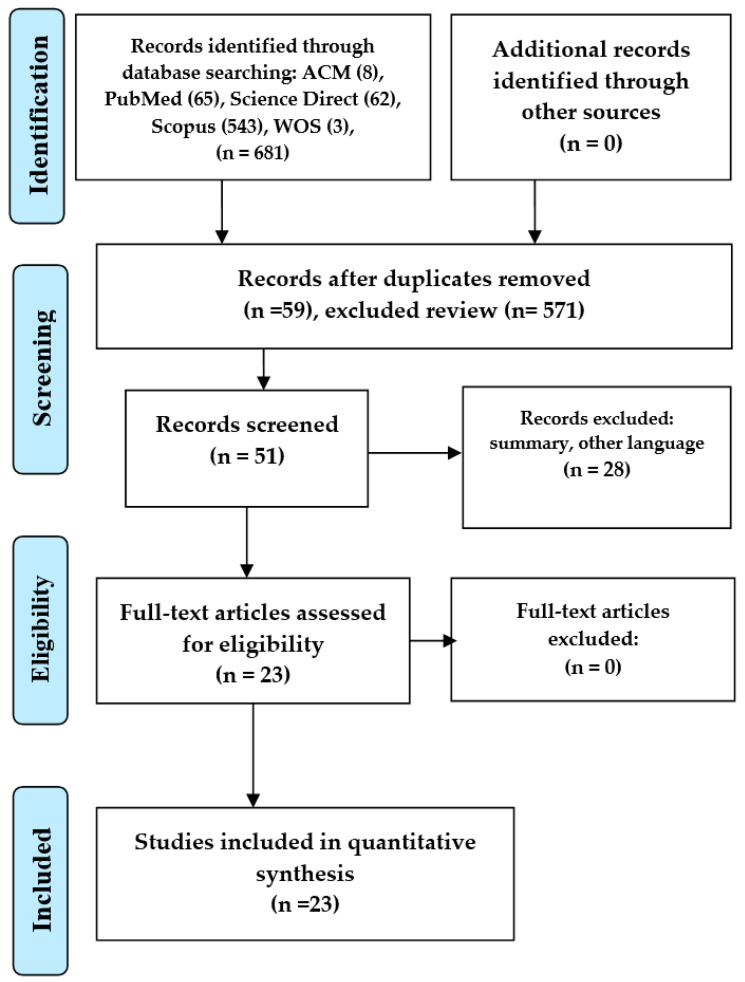
PRISMA flow diagram for the scoping review process.

**Figure 5 sensors-23-05048-f005:**
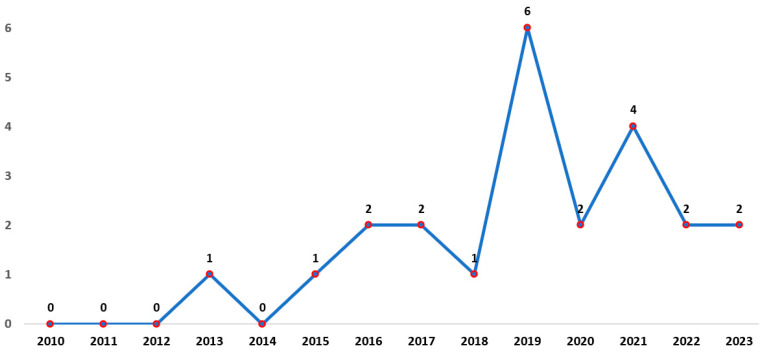
Articles published from 2010 to 2023.

**Figure 6 sensors-23-05048-f006:**
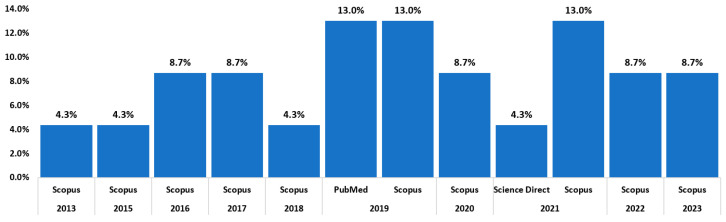
Documents by year based on when they are found.

**Figure 7 sensors-23-05048-f007:**
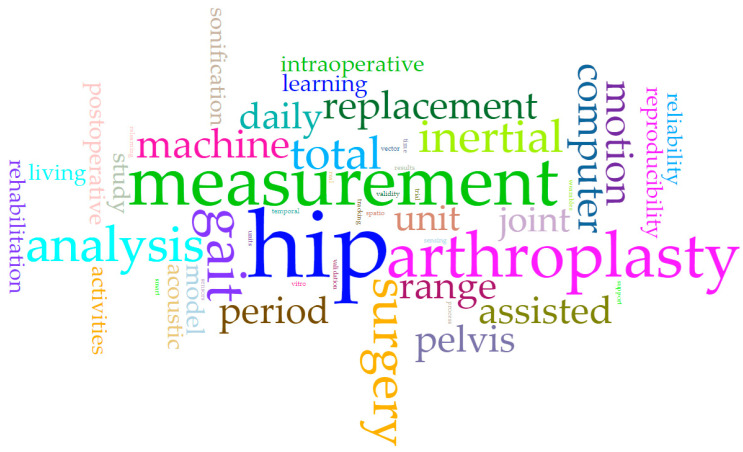
A word cloud of most frequently repeated keywords.

**Table 1 sensors-23-05048-t001:** Document quality assessment checklist.

N°	Quality Assessment Questions	Answer
QA1	Does the paper describe inertial sensors applied in hip arthroplasty?	(+1) Yes/(+0) No
QA2	Does the paper specify the evaluation methods applied to inertial sensors?	(+1) Yes/(+0) No
QA3	Does the paper discuss any findings of inertial sensors for hip arthroplasty evaluation?	(+1) Yes/(+0) No
QA4	Are limitations described in the inertial sensors considered for hip arthroplasty treatment?	(+1) Yes/(+0) No
QA5	Is the journal or conference in which the paper was published indexed in SJR?	(+1) if it is ranked Q1, (+0.75) if it is ranked Q2, (+0.50) if it is ranked Q3, (+0.25) if it is ranked Q4, (+0.0) if it is not ranked.

**Table 2 sensors-23-05048-t002:** Applied query string.

Database	String Search	Number of Studies
ACM Digital Library	[All: telerehabilitation and sensors] AND [All: inertial] AND [All: arthroplasty] AND [All: hip]	8
PubMed	telerehabilitation OR (sensors AND inertial) AND arthroplasty AND hip	65
ScienceDirect	(Telerehabilitation OR web) and (sensors and inertial) and arthroplasty and hip	62
Scopus	TITLE-ABS-KEY (web OR telerehabilitation OR telerehabilitation OR (sensors AND inertial) AND arthroplasty AND hip)	543
Web of Science	(((TS = (web)) OR TS = (telerehabilitation)) AND TS = (sensors AND inertial)) AND TS = (arthroplasty AND hip)	3
	Total number of studies	681

**Table 3 sensors-23-05048-t003:** Selected scientific articles and quality assessment outcomes.

ID	Publication Name	Quality Assessment								
		Quartile	SJR Factor	QA1	QA2	QA3	QA4	QA5	Score	Normalization
TeW2021	Automated detection and explainability of pathological gait patterns using a one-class support vector machine trained on inertial measurement unit based gait data	Q1	0.75	1	1	1	1	1.00	5.00	1.0
TaH2019	Monitoring hip posture in total hip arthroplasty using an inertial measurement unit-based hip smart trial system: An in vitro validation experiment using a fixed pelvis model	Q1	0.8	1	1	1	1	1.00	5.00	1.0
TeW2019	Towards an inertial sensor-based wearable feedback system for patients after total hip arthroplasty: Validity and applicability for gait classification with gait kinematics-based features	Q2	0.56	1	1	1	1	0.75	4.75	0.8
MaU2019	Innovative Force-PRO device to measure force and implant position in total hip arthroplasty	Q1	0.72	1	1	1	1	1.00	5.00	1.0
ReJ2019	Dual Mode Gait Sonification for Rehabilitation After Unilateral Hip Arthroplasty	Q3	0.73	1	1	1	1	0.50	4.50	0.5
ZuR2019	Validation of inertial measurement units with optical tracking system in patients operated with Total hip arthroplasty	Q2	0.67	1	1	1	1	0.75	4.75	0.8
KaM2022	Functional assessment of total hip arthroplasty using inertial measurement units: Improvement in gait kinematics and association with patient-reported outcome measures	Q2	0.84	1	1	1	1	0.75	4.75	0.8
WeJ2022	Design of an Affordable, Modular Implant Device for Soft Tissue Tension Assessment and Range of Motion Tracking During Total Hip Arthroplasty	Q2	0.84	1	1	1	1	0.75	4.75	0.8
PfS2016	A cost-effective surgical navigation solution for periacetabular osteotomy (PAO) surgery	Q1	1.00	1	1	1	1	1.00	5	1
ChH2021	An IMU-Based Real-Time Measuring System for Acetabular Prosthesis Implant Angles in THR Surgeries	Q1	0.93	1	1	1	1	1.00	5	1
BrM2020	Concurrent validity and inter trial reliability of a single inertial measurement unit for spatial-temporal gait parameter analysis in patients with recent total hip or total knee arthroplasty	Q1	0.68	1	1	1	1	1.00	5	1
GrH2019	Does the femoral head size in hip arthroplasty influence lower body movements during squats, gait and stair walking? A clinical pilot study based on wearable motion sensors	Q2	0.56	1	1	1	1	0.75	4.75	0.75
CaZ2017	IMU-based Real-time Pose Measurement system for Anterior Pelvic Plane in Total Hip Replacement Surgeries	N/A	0.00	1	1	1	1	0	4	0
McR2013	Inertial sensor based method for identifying spherical joint center of rotation	Q1	0.75	1	1	1	1	1.00	5	1
ShM2023	Inertial Tracking System for Monitoring Dual Mobility Hip Implants In Vitro	Q1	0.80	1	1	1	1	1.00	5	1
DinC2020	Interpretability of input representations for gait classification in patients after total hip arthroplasty	Q2	0.56	1	1	1	1	0.75	4.75	0.75
ChH2018	Measurement System for Attitude of Anterior Pelvic Plane and Implantation of Prothesis in THR Surgery	Q1	1.18	1	1	1	1	1.00	5	1
AlJ2021	Monitoring the rehabilitation progress using a DCNN and kinematic data for digital healthcare	N/A	0.00	1	1	1	1	0	4	0
SuS2017	Monocular Vision-and IMU-Based System for Prosthesis Pose Estimation during Total Hip Replacement Surgery	Q1	1.75	1	1	1	1	1.00	5	1
CaZ2016	Pose measurement of Anterior Pelvic Plane based on inertial measurement unit in total hip replacement surgeries	N/A	0.00	1	1	1	1	0	4	0
VaS2023	Repeatability of Inertial Measurement Units for Measuring Pelvic Mobility in Patients Undergoing Total Hip Arthroplasty	Q1	0.80	1	1	1	1	1.00	5	1
SuS2015	Smart trail with camera and inertial measurement unit for intraoperative estimation of hip range of motion in total hip replacement surgery	N/A	0.00	1	1	1	1	0	4	0
ZhW2021	Towards rehabilitation at home after total knee replacement	Q1	1.00	1	1	1	1	1.00	5	1

**Table 4 sensors-23-05048-t004:** Scientific articles were selected in this review.

N°	ID	Scientific Articles	Reference	Year
1	ShM2023	Inertial Tracking System for Monitoring Dual Mobility Hip Implants In Vitro	[19]	2023
2	VaS2023	Repeatability of Inertial Measurement Units for Measuring Pelvic Mobility in Patients Undergoing Total Hip Arthroplasty	[20]	2023
3	KaM2022	Functional assessment of total hip arthroplasty using inertial measurement units: Improvement in gait kinematics and association with patient-reported outcome measures	[21]	2022
4	WeJ2022	Design of an Affordable, Modular Implant Device for Soft Tissue Tension Assessment and Range of Motion Tracking During Total Hip Arthroplasty	[22]	2022
5	TeW2021	Automated detection and explainability of pathological gait patterns using a one-class support vector machine trained on inertial measurement unit based gait data	[23]	2021
6	ChH2021	An IMU-Based Real-Time Measuring System for Acetabular Prosthesis Implant Angles in THR Surgeries	[24]	2021
7	AlJ2021	Monitoring the rehabilitation progress using a DCNN and kinematic data for digital healthcare	[25]	2021
8	ZhW2021	Towards rehabilitation at home after total knee replacement	[26]	2021
9	BrM2020	Concurrent validity and inter trial reliability of a single inertial measurement unit for spatial-temporal gait parameter analysis in patients with recent total hip or total knee arthroplasty	[27]	2020
10	DinC2020	Interpretability of input representations for gait classification in patients after total hip arthroplasty	[28]	2020
11	TaH2019	Monitoring hip posture in total hip arthroplasty using an inertial measurement unit-based hip smart trial system: An in vitro validation experiment using a fixed pelvis model	[8]	2019
12	TeW2019	Towards an inertial sensor-based wearable feedback system for patients after total hip arthroplasty: Validity and applicability for gait classification with gait kinematics-based features	[9]	2019
13	MaU2019	Innovative Force-PRO device to measure force and implant position in total hip arthroplasty	[29]	2019
14	ReJ2019	Dual Mode Gait Sonification for Rehabilitation After Unilateral Hip Arthroplasty	[12]	2019
15	ZuR2019	Validation of inertial measurement units with optical tracking system in patients operated with Total hip arthroplasty	[13]	2019
16	GrH2019	Does the femoral head size in hip arthroplasty influence lower body movements during squats, gait and stair walking? A clinical pilot study based on wearable motion sensors	[30]	2019
17	ChH2018	Measurement System for Attitude of Anterior Pelvic Plane and Implantation of Prothesis in THR Surgery	[31]	2018
18	CaZ2017	IMU-based Real-time Pose Measurement system for Anterior Pelvic Plane in Total Hip Replacement Surgeries	[32]	2017
19	SuS2017	Monocular Vision-and IMU-Based System for Prosthesis Pose Estimation during Total Hip Replacement Surgery	[33]	2017
20	PfS2016	A cost-effective surgical navigation solution for periacetabular osteotomy (PAO) surgery	[34]	2016
21	CaZ2016	Pose measurement of Anterior Pelvic Plane based on inertial measurement unit in total hip replacement surgeries	[35]	2016
22	SuS2015	Smart trail with camera and inertial measurement unit for intraoperative estimation of hip range of motion in total hip replacement surgery	[36]	2015
23	McR2013	Inertial sensor based method for identifying spherical joint center of rotation	[37]	2013

**Table 5 sensors-23-05048-t005:** Results of the primary studies and answers to the research questions.

N°	ID	Ref	Sensor Type	Main Functions	Methods Used	Research	Advantages	Limitations
1	ShM2023	[19]	IMU	Dual mobility hip implants tracked.	Accurate inertial tracking of dual mobility hip implants.	Inertial sensors monitor hip arthroplasty.	High accuracy, real-time, non-invasive, portable, and accessible.	Drift, magnetic, calibrate, in vitro, in vivo
2	VaS2023	[20]	IMU	IMU repeatability in hip replacement.	IMU is reliable for measuring hip mobility.	IMU is useful for assessing THA.	Non-invasive, portable, real-time, accurate, and reliable.	Inaccurate, drift, noise, interference, placement.
3	KaM2022	[21]	IMU	Measure implant motion and position.	Data analyzed for post-surgery recovery.	Sensors produce sound for gait.	Accurate, portable, versatile, efficient, rehabilitation	Accuracy, stability, range, orientation, and cost.
4	WeJ2022	[22]	IMU	Motion aspects, velocity, acceleration, orientation.	A hip testing system for evaluation.	Monitoring, Evaluation, Rehabilitation, Arthroplasty, IMU.	Non-invasive, accurate, portable, real-time, feedback.	Reduce noise, bias, and data.
5	TeW2021	[23]	IMU	Measures implant angles in real-time.	We are detecting gait patterns with SVM.	Develop a single-class SVM algorithm.	Real-time gait analysis visualization.	Inaccurate joint angle measurement.
6	ChH2021	[24]	IMU	Real-time prosthesis angle measurement.	Real-time implantation angle measurement.	Improve precision, efficiency, angles, quantify results, and attention.	Monitor, measure, real-time, acetabular, precision	Accuracy, drift, calibration, motion, interference.
7	AlJ2021	[25]	IMU	Rehabilitation monitoring using DCNN.	Accurate gait classification, real-time feedback, and monitoring for rehabilitation	Improve accuracy, efficiency, measurements, outcomes, rehabilitation.	Improve accuracy, effectiveness, cost, and complexity.	Careful calibration, synchronization errors, combined measures, DCNNs, and training data.
8	ZhW2021	[26]	IMU	Rehabilitation post-knee surgery, remote training.	Approaches and technologies for remote rehabilitation.	Inertial sensors, monitor, TKR, rehabilitation, personalized.	Improve effectiveness, accessibility, feedback, engagement, and compliance.	The difficulty, calibration, accuracy, comfort, and inconvenience.
9	BrM2020	[27]	IMU	IMU gait analysis post-surgery.	IMU validated for gait measurement.	Validation, IMU, gait, parameters, implications	Convenient, efficient, accurate monitoring.	Choose wisely for gait analysis.
10	DinC2020	[28]	IMU	Gait classification input representations compared.	Important gait characteristics identified for classification.	Inertial sensors: gait assessment, hip arthroplasty, analysis data, clinical application, insights.	Transparency, interpretability, reliability, clinical, XAI.	Accurate placement, calibration, drift, errors, complex.
11	TaH2019	[8]	IMU	Motion, velocity, acceleration, orientation.	An intelligent hip testing system evaluates movements.	IMU system for hip rehabilitation.	Non-invasive, reliable, portable, real-time feedback.	Noise reduction and bias correction.
12	TeW2019	[9]	IMU	We are measuring gait kinematics, classification, and balance.	Classification of gait using gait characteristics, support time, and speed.	Applicable inertial sensors for gait classification.	Small, lightweight, inexpensive, easy, and valuable.	Periodic calibration for accuracy.
13	MaU2019	[29]	Force-PRO IMU	We are measuring force and implant motion.	IMU is reliable for post-arthroplasty posture.	Implant, rehabilitate load, movement, and arthroplasty.	Accurate, real-time, measurement, decision-making, non-invasive	Complex motion activity measurement.
14	ReJ2019	[12]	IMU	Implant motion tracking in 3D.	We are measuring gait after surgery, using sonification to improve gait.	Sound, sensor, movement, patient, improvement.	Accurate, portable, versatile, efficient gait.	Limited accuracy, stability, range, orientation, and cost.
15	ZuR2019	[13]	IMU	Analyze motion for abnormal deviations.	Optical data, inertial sensors, angles of movement, cross validation, accuracy.	Accurate, reliable hip arthroplasty movement.	Accurate, inertial, flexion, extension, adduction.	Accuracy, processing, sensors, movements, dependence
16	GrH2019	[30]	IMU	Hip replacement affects body movements.	Wearable sensors track lower body movements.	Hip arthroplasty: range, motion, sensors, functional, squatting.	Non-invasive, portable, cost-effective, objective, and affordable.	They have limited, small samples, variability, mechanics, and sensors.
17	ChH2018	[31]	IMU	Measurement system tracks pelvic plane during surgery.	Precise measurement of prosthesis position.	Inertial sensors, orthopedic surgery, accuracy, outcomes, development.	Accuracy, safety, cost, complexity, measurement	Inaccurate, external, magnetic, placement, obstruction.
18	CaZ2017	[32]	IMU	Real-time IMU measures pelvic position.	The real-time IMU system measures the iliac bone to provide information on the anterior pelvic plane position.	IMUs improve THR surgery accuracy.	Real-time, non-invasive, accurate, cost-effective, and practical.	Inaccurate, external magnetic field, placement, obstruction, optimization.
19	SuS2017	[33]	IMU	Measurement system for hip replacement using camera and IMU.	The proposed system refines position estimates using IMU and camera data.	Inertial sensors, monocular vision, prosthesis, surgery, accuracy.	Accuracy, safety, cost, complexity, improvement	Inaccurate, interference, temperature, humidity, limitation.
20	PfS2016	[34]	IMU	Open-source surgical navigation using optical tracking.	Cost-effective PAO navigation solution.	Improve precision, safety, results, quantity, and rehabilitation.	Accuracy, safety, real-time, tracking, precision.	Accuracy, drift, calibration, motion, interference.
21	CaZ2016	[35]	IMU	Improved hip arthroplasty outcomes using IMUs.	Kalman filter improves sensor accuracy.	Improve accuracy, safety, results, objective evaluations, and rehabilitation.	Easy, accurate, portable, affordable, and useful.	Errors, external interference, accuracy, position, placement.
22	SuS2015	[36]	IMU	“Smart Trail” estimates hip motion.	“Smart Trail” system estimates hip motion.	Improve surgical precision.	Improve accuracy, safety, cost, complexity, and motion.	Inertial sensor limitations: sensitivity, drift, position, calibration, sample.
23	McR2013	[37]	IMU	Method to locate ball joint’s center.	Inertial sensors estimate the joint center. They were validated with an optical measurement system.	Analyze kinematics, hip arthroplasty, inertial sensors, accuracy, and reliability.	Identify the center of rotation with sensors.	Inaccurate, time-consuming, impractical, spherical assumptions.

## Data Availability

Not applicable.

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
