# Peer review of "Inertial Sensors for Hip Arthroplasty Rehabilitation: A Scoping Review"

_sensors, 2023, doi:10.3390/s23115048_

Round 1
Reviewer 1 Report
The Document “Inertial Sensors for Hip Arthroplasty Rehabilitation: A Scoping Review" aims to provide a state of the art in the use of IMUs for postarthroplasty rehabilitation of the hip.
The methodology chosen is to conduct a systematic review, to identify major trends in the use and limitation of this measurement system.
While the work required to explore and analyze the literature data is considerable, the document in its current form is not ready for publication.
The introduction is confused; if the focus of the study is the use of IMUs for post-hip reeducation, the relevance or innovation of this technique or technology must be explained in relation to that already used and, above all, why it is not necessary to carry out this systematic review. This only appears in section 2 background and motivation. In the same section, references to knee replacement surgery are also provided that are not relevant. Furthermore, the FIG. 2 is not explained from a methodological point of view. Regarding the selection of research questions in paragraph 3.1, what is their relevance? , has this been assessed by an expert panel? Thus, the justification for the term "telerehabilitation" as a keyword must be subject to explanation. Moreover, it is also known that there is a great heterogeneity of the intrinsic performance of the IMUs in terms of noise and measurement drift, but also of the exploitation of measurements with or without data fusion. This point is not addressed by the authors. Validation procedures as well as the specificity and sensitivity of the chosen parameters are also ignored. The discussion is in narrative style that does not allow to make a synthesis. Finally, the authors are expected to propose a guideline on the use of IMUs for this specific application of hip replacement surgery. The conclusion should allow for the display of the eligibility criteria for UMI to become a medical reference device for the clinical monitoring of rehabilitation .
Author Response
Reviewer 1
The Document "Inertial Sensors for Hip Arthroplasty Rehabilitation: A Scoping Review" aims to provide a state of the art in the use of IMUs for postarthroplasty rehabilitation of the hip.
The methodology chosen is to conduct a systematic review, to identify major trends in the use and limitation of this measurement system.
While the work required to explore and analyze the literature data is considerable, the document in its current form is not ready for publication.
Response to Reviewer 1
Dear reviewer, thank you for the valuable comments and for taking the time to review our paper. Thank you for sharing the comment on the paper "Inertial sensors for hip arthroplasty rehabilitation: a scoping review." We understand your point of view, which is very important and allows us to improve the paper before publishing it.
The introduction is confused; if the focus of the study is the use of IMUs for post-hip reeducation, the relevance or innovation of this technique or technology must be explained in relation to that already used and, above all, why it is not necessary to carry out this systematic review. This only appears in section 2 background and motivation. In the same section, references to knee replacement surgery are also provided that are not relevant. Furthermore, the FIG. 2 is not explained from a methodological point of view.
Dear reviewer, thank you for your comments; we have improved the document with your comments; we added the relevance of this technology and justified the need for this review. We also remove information that is not relevant. In Figure 2, we detail the explanation from the methodological point of view. As shown in the article highlighted in yellow
Figure 2 shows a bibliometric review considered in this study with VOSviewer [7], a computer program used for bibliometric analysis. The review focused on identifying and analyzing the terms related to the technologies used and inertial sensors for measuring and monitoring the implant provided in arthroplasty treatments. The resulting graph shows the co-occurrence of these terms in the scientific literature, with larger nodes indicating more frequently occurring times and stronger relationships between the words.
It is important to note that the limited amount of scientific literature on these developments influenced the results obtained in the bibliometric analysis. The lack of data on this topic may also limit the generalizability of the findings and the ability to draw broader conclusions. Nonetheless, this bibliometric review provides valuable insight into the current research on using inertial sensors in arthroplasty treatments and can help guide future research in this area.
.
Regarding the selection of research questions in paragraph 3.1, what is their relevance? , has this been assessed by an expert panel? Thus, the justification for the term "telerehabilitation" as a keyword must be subject to explanation.
Dear reviewer, thank you for your comment; in section 3.1, we explain the relevance, the same one evaluated by a panel of experts.
As shown in the article highlighted in yellow
The research question is relevant as it addresses a topic of great importance in post-arthroplasty hip rehabilitation. Using inertial sensors in post-arthroplasty hip rehabilitation may provide a noninvasive and accurate way to monitor patient movement and position. This research may help develop personalized therapies and improve treatment efficacy.
In addition, the research question also addresses the limitations of inertial sensors, which may help researchers and medical professionals identify areas of improvement and develop solutions to overcome these limitations; from this question related to inertial sensors, we present three objectives.
The term telerehabilitation was included because, in future projects, we will apply this technology in a telerehabilitation system via the web; this idea was added as part of our future work.
In future projects, we will apply inertial sensor technology for personalized patient therapy in a web-based telerehabilitation system.
Moreover, it is also known that there is a great heterogeneity of the intrinsic performance of the IMUs in terms of noise and measurement drift, but also of the exploitation of measurements with or without data fusion. This point is not addressed by the authors. Validation procedures as well as the specificity and sensitivity of the chosen parameters are also ignored. The discussion is in narrative style that does not allow to make a synthesis.
Dear reviewer, thank you very much for your point of view and suggestions.
Concerning the heterogeneity of the intrinsic performance of inertial sensors, it refers to the variability in the precision and accuracy of the data obtained by the sensors in different conditions and environments. This variability may be due to noise and measurement drift specified in Table 5 in the "Advantages" and "Limitations" sections.
Exploiting measurements with or without data fusion refers to processing the data collected by inertial sensors. Inertial sensors can be used alone or with other types, such as optical motion sensors, to provide more accurate and comprehensive measurements. This issue is summarized in Table 5 in the "Methods used" section. In addition, we present some validation procedures and essential factors to consider in evaluating the specificity and sensitivity of the parameters chosen for applying inertial sensors in post-hip arthroplasty rehabilitation. As the cross-validation used to evaluate the accuracy of the prediction models based on the data collected by the inertial sensors they are detailed in Table 5 in "Main functions" and "Research".
The analysis of data obtained by the inertial sensors will be considered in future studies since they must be analyzed more profoundly to obtain helpful information about the position and movement of the patient. This process may include using time series analysis techniques and statistical analysis to identify patterns and trends in the data and may be addressed in a future topic with a more in-depth review applying meta-analysis.
Finally, the authors are expected to propose a guideline on the use of IMUs for this specific application of hip replacement surgery. The conclusion should allow for the display of the eligibility criteria for UMI to become a medical reference device for the clinical monitoring of rehabilitation.
Dear reviewer, thank you very much for your suggestions.
Dear reviewer, we have applied what was suggested. We incorporated into our document a guide on the use of IMUs for this specific application of hip replacement surgery. As evidenced in the document highlighted in yellow. In the discussion section, such as you can provide evidence.
The authors propose a guideline on using inertial sensors (IMU) for hip arthroplasty rehabilitation: Selecting suitable inertial sensors is essential to ensure accurate and reliable measurements. Arbitration, sampling rate, size, and battery life should be considered when selecting inertial sensors.
The inertial sensors must be placed correctly and securely on the patient to ensure accurate and reliable measurements. Sensors should be securely attached to the patient and placed strategically to capture relevant hip position and motion.
They are assessing the quality of measurements obtained by inertial sensors. Researchers and medical professionals must assess the measurements' accuracy, repeatability, and validity to ensure their reliability.
The data processed by inertial sensors must be processed appropriately to obtain helpful information about the patient's movement and position. This process may include the fusion of data from different types of sensors to provide a more complete and accurate measurement. It can be used to develop personalized therapies for the patient. Medical professionals can use the data to tailor treatment to patient's needs and assess rehabilitation progress.
Progress evaluation to assess the patient's progress in hip rehabilitation after arthroplasty. Researchers and medical professionals can use the data to assess the efficacy and adjust the treatment plan. Patient safety; ensure that using inertial sensors is safe for the patient. Sensors must be correctly and securely placed to avoid injury or discomfort to the patient.

Reviewer 2 Report
This manuscript reports inertial sensor for hip rehabilitation. The conclusion is reasonable. The following needs to polish before te final publications,
1, The background needs to rewrite. There are too many paragrams, the authors need to have clear mind on this section.
2, As to the inertial sensors, the author can cite more papers to support the background, such as Development trends and perspectives of future sensors and MEMS/NEMS. Micromachines. 11, 7, 541, 2020.
3, What is the nolverty of this manuscript? please make more comments on the advantage of the weakness of the inertial sensor.
4, The section 5 and 6 can be combine into one.
5, What the reason to make the evaluaton of Table 3?
Author Response
Reviewer 2
This manuscript reports inertial sensor for hip rehabilitation. The conclusion is reasonable. The following needs to polish before te final publications,
Dear reviewer, all the authors thank you for your valuable comments and the time invested in commenting on our document to improve it before its publication.
1, The background needs to rewrite. There are too many paragrams, the authors need to have clear mind on this section.
Dear reviewer, we have improved what you requested to reduce unnecessary paragraphs and explain it more clearly.
2, As to the inertial sensors, the author can cite more papers to support the background, such as Development trends and perspectives of future sensors and MEMS/NEMS. Micromachines. 11, 7, 541, 2020.
Dear reviewer, thank you very much for your recommendation. We have supported the background with the indicated reference, as evidenced in the document.
In this context, future sensors and systems development trends and prospects focus on typical MEMS sensors for Internet of Things applications. Future sensor trends focus on intelligence and lower power consumption, and [6] future sensors, such as event-based sensors, address big data and human-machine issues. Artificial intelligence and virtual reality technologies using sensor nodes and their wave identification are presented as future trends for various scenarios.
[6]Zhu J, Liu X, Shi Q, et al (2019) Development trends and perspectives of future sensors and MEMS/NEMS. Micromachines 11:7. https://doi.org/10.3390/mi11010007
3, What is the nolverty of this manuscript? please make more comments on the advantage of the weakness of the inertial sensor.
Dear reviewer, we have applied what was suggested and placed what is indicated in the discussion section, as can be seen by highlighting it in yellow.
Regarding the advantage of the weakness of the inertial sensor, it is essential to note that a weak inertial sensor may provide less accurate and reliable measurements compared to a robust inertial sensor. However, in some cases, this weakness can be an advantage. For example, a weak inertia sensor may better capture smooth and fluid movements than fast and jerky movements in human motion monitoring. In addition, a weak inertial sensor may be less expensive and lighter, making it more suitable for portable and long-term use applications.
4, The section 5 and 6 can be combine into one.
Thanks for the comments; we have left them as requested in the structure of the journal.
5, What the reason to make the evaluaton of Table 3?
Dear reviewer, thank you for your comments regarding your concern.
Table 3 presents the validation of the quality of the articles. Valuing the articles' quality is essential because it helps ensure that the information presented is reliable and accurate. By validating the quality of articles, possible biases and limitations in the methodology or results of the study can be identified. Table 3 provides a means to validate the quality of articles included in a study or review. The five quality questions and the score applied can help determine the validity and reliability of the study results and the quality of the methodological design and execution of the study. The normalization column uses a standard scale ranging from 0 to 1 to allow comparison and statistical analysis between the different studies included in the analysis. This table facilitates the interpretation of the results and allows a fair comparison between the different studies.

Round 2
Reviewer 2 Report
The authors did answer all my issues. I recomment to publish in the current form.